# Subset Selection and Summarization in Sequential Data

**Ehsan Elhamifar**
Computer and Information Science College
Northeastern University
Boston, MA 02115
eelhami@ccs.neu.edu

**M. Clara De Paolis Kaluza**
Computer and Information Science College
Northeastern University
Boston, MA 02115
clara@ccs.neu.edu

## Abstract

Subset selection, which is the task of finding a small subset of representative items from a large ground set, finds numerous applications in different areas. Sequential data, including time-series and ordered data, contain important structural relationships among items, imposed by underlying dynamic models of data, that should play a vital role in the selection of representatives. However, nearly all existing subset selection techniques ignore underlying dynamics of data and treat items independently, leading to incompatible sets of representatives. In this paper, we develop a new framework for sequential subset selection that finds a set of representatives compatible with the dynamic models of data. To do so, we equip items with transition dynamic models and pose the problem as an integer binary optimization over assignments of sequential items to representatives, that leads to high encoding, diversity and transition potentials. Our formulation generalizes the well-known facility location objective to deal with sequential data, incorporating transition dynamics among facilities. As the proposed formulation is non-convex, we derive a max-sum message passing algorithm to solve the problem efficiently. Experiments on synthetic and real data, including instructional video summarization, show that our sequential subset selection framework not only achieves better encoding and diversity than the state of the art, but also successfully incorporates dynamics of data, leading to compatible representatives.

## 1 Introduction

Subset selection is the task of finding a small subset of most informative items from a ground set. Besides helping to reduce the computational time and memory of algorithms, due to working on a much smaller representative set [1], it has found numerous applications, including, image and video summarization [2, 3, 4], speech and document summarization [5, 6, 7], clustering [8, 9, 10, 11, 12], feature and model selection [13, 14, 15, 16], sensor placement [17, 18], social network marketing [19] and product recommendation [20]. Compared to dictionary learning methods such as Kmeans [21], KSVD [22] and HMMs [23], that *learn* centers/atoms in the input-space, subset selection methods *choose* centers/atoms from the given set of items.

Sequential data, including time-series such as video, speech, audio and sensor measurements as well as ordered data such as text, form an important large part of modern datasets, requiring effective subset selection techniques. Such datasets contain important structural relationships among items, often imposed by underlying dynamic models, that should play a vital role in the selection of representatives. For example, there exists a logical way in which segments of a video or sentences of a document are connected together and treating segments/sentences as a bag of randomly permutable items results in losing the semantic content of the video/document. However, existing subset selection methods

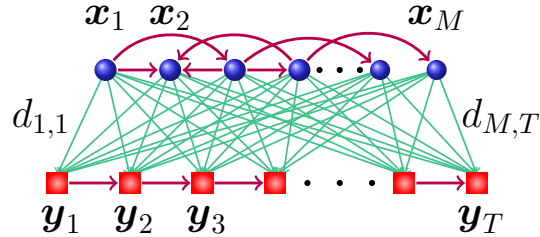

Figure 1: We propose a framework, based on a generalization of the facility location problem, for the summarization of sequential data. Given a source set of items $\{\boldsymbol{x}_1, \ldots, \boldsymbol{x}_M\}$ with a dynamic transition model and a target set of sequential items $(\boldsymbol{y}_1, \ldots, \boldsymbol{y}_T)$, we propose a framework to find a sequence of representatives from the source set that has a high global transition probability and well encodes the target set.

ignore these relationships and treat items independent from each other. Thus, there is a need for sequential subset selection methods that, instead of treating items independently, use the underlying dynamic models of data to select high-quality, diverse and compatible representatives.

**Prior Work:** A subset selection framework consists of three main components: i) the inputs to the algorithm; ii) the objective function to optimize, characterizing the informativeness and diversity of selected items; iii) the algorithm to optimize the objective function. The inputs to subset selection algorithms are in the form of either feature vector representations or pairwise similarities between items. Several subset selection criteria have been studied in the literature, including maximum cut objective [24, 25], maximum marginal relevance [26], capacitated and uncapacitated facility location objectives [27, 28], multi-linear coding [29, 30] and maximum volume subset [6, 31], which all try to characterize the informativeness/value of a subset of items in terms of ability to represent the entire distribution and/or having minimum information overlap among selected items. On the other hand, optimizing almost all subset selection criteria is, in general, NP-hard and non-convex [25, 32, 33, 34], which has motivated the development and study of approximate methods for optimizing these criteria. This includes greedy approximate algorithms [28] for maximizing submodular functions, such as graph-cuts and facility location, which have worst-case approximation guarantees, as well as sampling methods from Determinantal Point Process (DPP) [6, 31], a probability measure on the set of all subsets of a ground set, for approximately finding the maximum volume subset. Motivated by the maturity of convex optimization and advances in sparse and low-rank recovery, recent methods have focused on convex relaxation-based methods for subset selection [8, 9, 2, 35, 36].

When it comes to sequential data, however, the majority of subset selection methods ignore the underlying dynamics and relationships among items and treat items independent from each other. Recent results in [37, 3] have developed interesting extensions to DPP-based subset selection, by capturing representatives in a sequential order such that newly selected representatives are diverse with respect to the previously selected ones. However, sequential diversity by itself is generally insufficient, especially, when the sequence of diverse selected items are unlikely to follow each other according to underlying dynamic models. For example, in a video/document on a specific topic with intermediate irrelevant scenes/sentences to the topic, promoting sequential diversity results in selecting irrelevant scenes/sentences. [38] extends submodular functions to capture ordered preferences among items, where ordered preferences are represented by a directed acyclic graph over items, and presents a greedy algorithm to pick edges instead of items. The method, however, cannot deal with arbitrary graphs, such as Markov chains with cycles. On the other hand, while Hidden Markov Models (HMMs) [23, 39] and dynamical systems [40, 41] have been extensively studied for modeling sequential data, they have not been properly exploited in the context of subset selection.

**Paper Contributions:** In this paper we develop a new framework for sequential subset selection that incorporates the dynamic model of sequential data into subset selection. We develop a new class of objective functions that promotes to select not only high-quality and diverse items, but also a sequence of representatives that are compatible with the dynamic model of data. To do so, we propose a dynamic subset selection framework, where we equip items with transition probabilities and design objective functions to select representatives that well capture the data distribution with a high overall transition probability in the sequence of representatives, see Figure 1. Our formulation generalizes the facility location objective [27, 28] to sequential data, by incorporating transition dynamics among facilities. Since our proposed integer binary optimization is non-convex, we develop a max-sum

message passing framework to solve the problem efficiently. By experiments on synthetic and real data, including instructional video summarization, we show that our method outperforms the state of the art in terms of selecting representatives with better encoding, diversity and dynamic compatibility.

## 2 Subset Selection for Sequential Data

Sequential data, including time-series and ordered data contain important structural relationships among items, often imposed by underlying dynamic models of data, that should play a vital role in the selection of representatives. In this section, we develop a new framework for sequential subset selection that incorporates underlying dynamic models and relationships among items into subset selection. More specifically, we propose a dynamic subset selection framework, where we equip items with transition probabilities and design objectives to select representatives that capture the data distribution with a high transition probability in the sequence of representatives. In the next section, we develop an efficient algorithm to solve the proposed optimization problem.

### 2.1 Sequential Subset Selection Formulation

Assume we have a source set of items $\mathbb{X} = \{\boldsymbol{x}_1, \ldots, \boldsymbol{x}_M\}$, equipped with a transition model, $p(\boldsymbol{x}_{i'} | \boldsymbol{x}_{i_1}, \ldots, \boldsymbol{x}_{i_n})$, between items, and a target set of sequential items $\mathbb{Y} = (\boldsymbol{y}_1, \ldots, \boldsymbol{y}_T)$. Our goal is to find a small representative subset of $\mathbb{X}$ that well encode $\mathbb{Y}$, while the set of representatives are compatible according to the dynamic model of $\mathbb{X}$. Let $\boldsymbol{x}_{r_t}$ be the representative of $\boldsymbol{y}_t$ for $t \in \{1, \ldots, T\}$. We propose a potential function $\Psi(r_1, \ldots, r_T)$ whose maximization over all possible assignments $(r_1, \ldots, r_T) \subseteq \{1, \ldots, M\}^T$, i.e.,

$$\max_{(r_1,\ldots,r_T) \subseteq \{1,\ldots,M\}^T} \Psi(r_1, \ldots, r_T) \tag{1}$$

achieves the three goals of i) minimizing the encoding cost of $\mathbb{Y}$ via the representative set; ii) selecting a small set of representatives from $\mathbb{X}$; iii) selecting an ordered set of representatives $(\boldsymbol{x}_{r_1}, \ldots, \boldsymbol{x}_{r_T})$ that are compatible with the dynamics on $\mathbb{X}$.

To tackle the problem, we consider a decomposition of the potential function $\Psi$ into the product of three potentials, corresponding to the three aforementioned objectives, as

$$\Psi(r_1, \ldots, r_T) \triangleq \Phi_{\text{enc}}(r_1, \ldots, r_T) \times \Phi_{\text{card}}(r_1, \ldots, r_T) \times \Phi_{\text{dyn}}(r_1, \ldots, r_T), \tag{2}$$

where $\Phi_{\text{enc}}(r_1, \ldots, r_T)$ denotes the encoding potential that favors selecting a representative set from $\mathbb{X}$ that well encodes $\mathbb{Y}$, $\Phi_{\text{card}}(r_1, \ldots, r_T)$ denotes the cardinality potential that favors selecting a small number of distinct representatives. Finally, $\Phi_{\text{dyn}}(r_1, \ldots, r_T)$ denotes the dynamic potential that favors selecting an ordered set of representatives that are likely to be generated by the underlying dynamic model on $\mathbb{X}$. Next, we study each of the three potentials.

**Encoding Potential:** Since the encoding of each item of $\mathbb{Y}$ depends on its own representative, we assume that the encoding potential function factorizes as

$$\Phi_{\text{enc}}(r_1, \ldots, r_T) = \prod_{t=1}^{T} \phi_{\text{enc},t}(r_t), \tag{3}$$

where $\phi_{\text{enc},t}(i)$ characterizes how well $\boldsymbol{x}_i$ encodes $\boldsymbol{y}_t$ and becomes larger when $\boldsymbol{x}_i$ better represents $\boldsymbol{y}_t$. In this paper, we assume that $\phi_{\text{enc},t}(i) = \exp(-d_{i,t})$, where $d_{i,t}$ indicates the dissimilarity of $\boldsymbol{x}_i$ to $\boldsymbol{y}_t$.[1] A lower dissimilarity $d_{i,t}$ means that $\boldsymbol{x}_i$ better encodes/represents $\boldsymbol{y}_t$.

**Cardinality Potential:** Notice that maximizing the encoding potential alone results in selecting many representatives. Hence, we consider a cardinality potential to restrict the total number of representatives. Denoting the number of representatives by $|\{r_1, \ldots, r_T\}|$, we consider

$$\Phi_{\text{card}}(r_1, \ldots, r_T) = \exp(-\lambda |\{r_1, \ldots, r_T\}|), \tag{4}$$

which promotes to select a small number of representatives. The parameter $\lambda > 0$ controls the effect of the cardinality on the global potential $\Psi$, where a close to zero $\lambda$ ignores the effect of cardinality potential, resulting in many representatives, and a larger $\lambda$ results in a smaller representative set.

**Dynamic Potential:** While encoding and cardinality potentials together promote selecting a few representatives from $\mathbb{X}$ that well encode $\mathbb{Y}$, there is no guarantee that the sequence of representatives $(\boldsymbol{x}_{r_1}, \ldots, \boldsymbol{x}_{r_T})$ is compatible with the underlying dynamic of $\mathbb{X}$. Thus, we introduce a dynamic potential that measures the compatibility of the sequence of representatives. To do so, we consider an $n$-th order Markov Model to represent the dynamic relationships among the items in $\mathbb{X}$, where the selection of the representative $\boldsymbol{x}_{r_t}$ depends on the $m$ previously selected representatives, i.e., $\boldsymbol{x}_{r_{t-1}}, \ldots, \boldsymbol{x}_{r_{t-n}}$. More precisely, we consider

$$\Phi_{\text{dyn}}(r_1, \ldots, r_T) = \left( p_1(\boldsymbol{x}_{r_1}) \times \prod_{t=2}^{n} p_t(\boldsymbol{x}_{r_t} | \boldsymbol{x}_{r_{t-1}}, \ldots, \boldsymbol{x}_{r_1}) \times \prod_{t=n+1}^{T} p_t(\boldsymbol{x}_{r_t} | \boldsymbol{x}_{r_{t-1}}, \ldots, \boldsymbol{x}_{r_{t-n}}) \right)^{\beta}, \quad (5)$$

where $p_t(\boldsymbol{x}_i)$ indicates the probability of selecting $\boldsymbol{x}_i$ as the representative of $\boldsymbol{y}_t$ and $p_t(\boldsymbol{x}_{i'} | \boldsymbol{x}_{i_1}, \ldots, \boldsymbol{x}_{i_n})$ denotes the probability of selecting $\boldsymbol{x}_{i'}$ as the representative of $\boldsymbol{y}_t$ given that $\boldsymbol{x}_{i_1}, \ldots, \boldsymbol{x}_{i_n}$ has been selected as the representative of $\boldsymbol{y}_{t-1}, \ldots, \boldsymbol{y}_{t-n}$, respectively. The regularization parameter $\beta > 0$ determines the effect of the dynamic potential on the overall potential $\Psi$, where a close to zero $\beta$ results in discounting the effect of the dynamic of $\mathbb{X}$. As a result, maximizing the dynamic potential promotes to select a sequence of representatives that are highly likely to follow the dynamic model on the source set. In this paper, we assume that the transition dynamic model on the source set is given and known. In the experiments on video summarization, we learn the dynamic model by fitting a hidden Markov Model to data.

## 2.2 Optimization Framework for Sequential Subset Selection

In the rest of the paper, we consider a first order Markov model, which performs well in the application studied in the paper (our proposed optimization can be generalized to $n$-th order Markov models as well). Putting all three potentials together, we consider maximization of the global potential function

$$\Psi = \prod_{t=1}^{T} \phi_{\text{enc},t}(r_t) \times \Phi_{\text{card}}(r_1, \ldots, r_T) \times \left( p_1(\boldsymbol{x}_{r_1}) \times \prod_{t=2}^{T} p_t(\boldsymbol{x}_{r_t} | \boldsymbol{x}_{r_{t-1}}) \right)^{\beta}. \quad (6)$$

over all possible assignments $(r_1, \ldots, r_T) \subseteq \{1, \ldots, M\}^T$. To do so, we cast the problem as an integer binary optimization. We define binary assignment variables $\{z_{i,t}\}_{i=1,\ldots,M}^{t=1,\ldots,T}$, where $z_{i,t} \in \{0, 1\}$ indicates if $\boldsymbol{x}_i$ is a representative of $\boldsymbol{y}_t$. Since each item $\boldsymbol{y}_t$ is associated with only a single representative, we have $\sum_{i=1}^{M} z_{i,t} = 1$. Also, we define variables $\{\delta_i\}_{i=1,\ldots,M}$ and $\{u_{i',i}^t\}_{i,i'=1,\ldots,M}^{t=1,\ldots,T}$, where $\delta_i \in \{0, 1\}$ indicates if $\boldsymbol{x}_i$ is a representative of $\boldsymbol{y}_1$ and $u_{i',i}^t \in \{0, 1\}$ indicates if $\boldsymbol{x}_{i'}$ is a representative of $\boldsymbol{y}_t$ given that $\boldsymbol{x}_i$ is a representative of $\boldsymbol{y}_{t-1}$. As we will show, $\{\delta_i\}$ and $\{u_{i',i}^t\}$ are related to $\{z_{i,t}\}$, hence, the final optimization only depends on $\{z_{i,t}\}$.

Using the variables defined above, we can rewrite the global potential function in (6) as

$$\Psi = \prod_{t=1}^{T} \prod_{i=1}^{M} \phi_{\text{enc},t}(i)^{z_{i,t}} \times \Phi_{\text{card}}(r_1, \ldots, r_T) \times \prod_{i=1}^{M} p_1(\boldsymbol{x}_i)^{\beta \delta_i} \times \prod_{t=2}^{T} \prod_{i'=1}^{M} \prod_{i=1}^{M} p_t(\boldsymbol{x}_{i'} | \boldsymbol{x}_i)^{\beta u_{i',i}^t}. \quad (7)$$

We can equivalently maximize the logarithm of $\Psi$, which is to maximize

$$\sum_{t=1}^{T} \sum_{i=1}^{M} -z_{i,t} d_{i,t} + \log \Phi_{\text{card}}(r_1, \ldots, r_T) + \sum_{i=1}^{M} \delta_i \log p_1(\boldsymbol{x}_i) + \sum_{t=2}^{T} \sum_{i,i'=1}^{M} u_{i',i} \log p_t(\boldsymbol{x}_{i'} | \boldsymbol{x}_i), \quad (8)$$

where we used $\log \phi_{\text{enc},t}(i) = -d_{i,t}$. Notice that $\{\delta_i\}$ and $\{u_{i',i}^t\}$ can be written as functions of the assignment variables $\{z_{i,t}\}$. Denoting the indicator function by $1_{(\cdot)}$, which is one when its argument is true and is zero otherwise, we can write $\delta_i = 1_{(r_1=i)}$ and $u_{i',i}^t = 1_{(r_t=i', r_{t-1}=i)}$. Hence, we have

$$\delta_i = z_{i,1}, \quad u_{i',i}^t = z_{i,t-1} z_{i',t}. \quad (9)$$

As a result, we can rewrite the maximization in (8) as the equivalent optimization

$$\max_{\{z_{i,t}\}} \quad \sum_{t=1}^{T} \sum_{i=1}^{M} -z_{i,t} d_{i,t} + \log \Phi_{\text{card}}(r_1, \ldots, r_T) + \beta (\sum_{i=1}^{M} z_{i,1} \log p_1(\boldsymbol{x}_i)$$

$$+ \sum_{t=2}^{T} \sum_{i,i'=1}^{M} z_{i,t-1} z_{i',t} \log p_t(\boldsymbol{x}_{i'} | \boldsymbol{x}_i)) \quad \text{s. t.} \quad z_{i,t} \in \{0, 1\}, \quad \sum_{i=1}^{M} z_{i,t} = 1, \forall i, t. \quad (10)$$

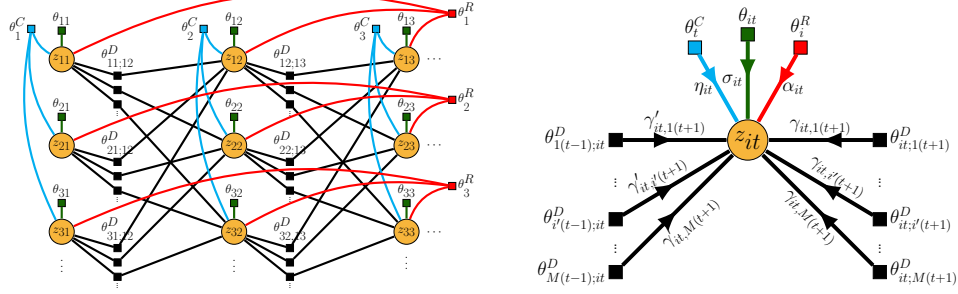

Figure 2: Left: Factor graph representing (12). Right: Messages from each factor to a variable node $z_{i,t}$.

It is important to note that if $\boldsymbol{x}_i$ becomes a representative of some items in $\mathbb{Y}$, then $\|\,[\,z_{i,1}\ \cdots\ z_{i,T}]\,\|_\infty$ would be 1. Hence, the number of representatives is given by $\sum_{i=1}^{M}\|\,[\,z_{i,1}\ \cdots\ z_{i,T}]\,\|_\infty$. As a result, we can rewrite the cardinality potential in (4) as

$$\Phi_{\mathrm{card}}(r_1,\ldots,r_T) = \exp(-\lambda\sum_{i=1}^{M}\|\,[\,z_{i,1}\ \cdots\ z_{i,T}]\,\|_\infty). \tag{11}$$

Finally, considering a homogeneous Markov Model on the dynamics of the source set, where $p_t(\cdot|\cdot) = p(\cdot|\cdot)$, i.e., transitioning from $\boldsymbol{x}_i$ as the representative of $\boldsymbol{y}_{t-1}$ to $\boldsymbol{x}_{i'}$ as the representative of $\boldsymbol{y}_t$ does not depend on $t$, we propose to solve the optimization

$$\begin{aligned}
\max_{\{z_{i,t}\}}\ &\sum_{t=1}^{T}\sum_{i=1}^{M} -z_{i,t}d_{i,t} - \lambda\sum_{i=1}^{M}\|\,[\,z_{i,1}\ \cdots\ z_{i,T}]\,\|_\infty + \beta\Big(\sum_{i=1}^{M} z_{i,1}\log p_1(\boldsymbol{x}_i)\\
&+ \sum_{t=2}^{T}\sum_{i,i'=1}^{M} z_{i,t-1}z_{i',t}\log p(\boldsymbol{x}_{i'}|\boldsymbol{x}_i)\Big)\quad\text{s.t.}\quad z_{i,t}\in\{0,1\},\ \ \sum_{i=1}^{M} z_{i,t} = 1,\ \forall\,i,t.
\end{aligned} \tag{12}$$

In our proposed formulation above, we assume that the dissimilarities $\{d_{i,t}\}$ and the dynamic models, i.e., the probabilities $p_1(\cdot)$ and $p(\cdot|\cdot)$, are known. These models can be given by prior knowledge or by learning from training data, as we show in the experiments. It is important to notice that the optimization in (12) is non-convex, due to binary optimization variables and quadratic terms in the objective function, which is not necessarily positive semi-definite (this can be easily seen when $p(x_{i'}|x_i)\neq p(x_i|x_{i'})$ for some $i,i'$). In the next section, we treat (12) as a MAP inference on binary random variables and develop a message passing algorithm to find the hidden values $\{z_{i,t}\}$.

Once we solve the optimization in (12), we can obtain the representatives as the items of $\mathbb{X}$ for which $z_{i,t}$ is non-zero for some $t$. Moreover, we can obtain the segmentation of the sequential items in $\mathbb{Y}$ according to their assignments to the representatives. In fact, the sequence of representatives obtained by our proposed optimization in (12) not only corresponds to diverse items that well encode the sequential target data, but also is compatible with the underlying dynamic of the source data.

**Remark 1** *Without the dynamic potential, i.e., with $\beta = 0$, our proposed optimization in (12) reduces to the uncapacitated facility location objective. Hence, our framework generalizes the facility location to sequential data by considering transition dynamics among facilities (source set items). On the other hand, if we assume uniform distributions for the initial and transition probabilities, the dynamic term (last term) in our objective function becomes a constant, hence, our formulation reduces to the uncapacitated facility location. As a result, our framework generalizes the facility location, where we consider arbitrary initial and transition probabilities on $\mathbb{X}$ instead of a uniform distribution.*

## 3 Message Passing for Sequential Subset Selection

In this section, we develop an efficient message passing algorithm to solve the proposed optimization in (12). To do so, we treat the sequential subset selection as a MAP inference, where $\{z_{i,t}\}$ correspond to binary random variables whose joint log-likelihood is given by the objective function in (12). We represent the log-likelihood, i.e., the objective function in (12), with a factor graph [42], which is shown in Figure 2. Recall that a factor graph is a bipartite graph that consists of variable nodes and

factor nodes, where every factor evaluates a potential function over variables it is connected to. The log-likelihood is then proportional to the sum of all factor potentials.

To form the factors corresponding to the objective function in (12), we define $m_{i,i'} \triangleq \log p(\boldsymbol{x}_{i'}|\boldsymbol{x}_i)$ and $\bar{d}_{i,t} \triangleq d_{i,t} - \log p_1(\boldsymbol{x}_i)$ if $t = 1$ and $\bar{d}_{i,t} \triangleq d_{i,t}$ for all other values of $t$. Denoting $\boldsymbol{z}_{i,:} \triangleq [z_{i,1} \;\; \cdots \;\; z_{i,T}]^\top$ and $\boldsymbol{z}_{:,t} \triangleq [z_{1,t} \;\; \cdots \;\; z_{M,t}]^\top$, we define factor potentials corresponding to our framework, shown in Figure 2. More specifically, we define the encoding and dynamic potentials, respectively, as $\theta_{i,t}(z_{i,t}) \triangleq -\bar{d}_{i,t} z_{i,t}$ and $\theta_{i,t-1;i',t}^D(z_{i,t-1}, z_{i',t}) \triangleq m_{i,i'} z_{i,t-1} z_{i',t}$. Moreover we define the cardinality and constraint potentials, respectively, as

$$\theta_i^R(\boldsymbol{z}_{i,:}) \triangleq \begin{cases} -\lambda, & \|\boldsymbol{z}_{i,:}\|_\infty > 0 \\ 0, & \text{otherwise} \end{cases}, \quad \theta_t^C(\boldsymbol{z}_{:,t}) \triangleq \begin{cases} 0, & \sum_{i=1}^M z_{i,t} = 1 \\ -\infty, & \text{otherwise} \end{cases}.$$

The MAP formulation of our sequential subset selection is then given by

$$\max_{\{z_{i,t}\}} \sum_{t=1}^T \sum_{i=1}^M \theta_{i,t}(z_{i,t}) + \sum_{i=1}^M \theta_i^R(\boldsymbol{z}_{i,:}) + \sum_{t=1}^T \theta_l^C(\boldsymbol{z}_{:,t}) + \beta \sum_{t=1}^{T-1} \sum_{i'=1}^M \sum_{i=1}^M \theta_{i,t-1;i',t}^D(z_{i,t-1}, z_{i',t}). \quad (13)$$

To perform MAP inference, we use the max-sum message passing algorithm, which iteratively updates messages between variable and factor nodes in the graph. In our framework, the incoming messages to each variable node $z_{i,t}$ are illustrated in Figure 2. Messages are computed as follows (please see the supplementary materials for the derivations).

$$\sigma_{i,t} \leftarrow -\bar{d}_{i,t} \quad (14)$$

$$\gamma_{i,t;j,t+1} \leftarrow \max\{0, \, m_{i,j} + \rho\} - \max\{0, \, \rho\} \quad (15)$$

$$\gamma'_{i,t-1;j,t} \leftarrow \max\{0, \, m_{i,j} + \rho'\} - \max\{0, \, \rho'\} \quad (16)$$

$$\eta_{i,t} \leftarrow -\max_{k \neq i',i}\left\{\alpha_{i',1} - \bar{d}_{i',1} + \sum_{j=1}^M \gamma_{i',t;j,t+1} + \sum_{j=1}^M \gamma'_{j,t-1;i',t}\right\} \quad (17)$$

$$\alpha_{i,t} \leftarrow \min\left\{0, -\lambda + \sum_{k \neq t} \max\left\{0, -\bar{d}_{i,k} + \eta_{i,k} + \sum_{j=1}^M (\gamma_{i,k;j,k+1} + \gamma'_{j,k-1;i,k})\right\}\right\} \quad (18)$$

where, for brevity of notation, we have defined $\rho$ and $\rho'$ as

$$\rho \overset{\triangle}{=} -\bar{d}_{j,t+1} + \alpha_{j,t+1} + \eta_{j,t+1} + \sum_{k=1}^M \gamma_{j,t+1;k,t+2} + \sum_{k \neq i} \gamma'_{k,t;j,t+1}, \quad (19)$$

$$\rho' \overset{\triangle}{=} -\bar{d}_{i,t-1} + \alpha_{i,t-1} + \eta_{i,t-1} + \sum_{k \neq i} \gamma_{i,t-1;k,t} + \sum_{k=1}^M \gamma'_{k,t-2;i,t-1}. \quad (20)$$

The update of messages continues until convergence, when each variable $z_{i,t}$ is assigned to the value that maximizes the sum of its incoming messages. It is important to note that the max-sum algorithm always converges to the optimal MAP assignment on trees, and has shown good performance on graphs with cycles in many applications, including our work. We also use a dampening factor $\lambda \in [0, 1)$ on message updates as so that a message $\mu$ is computed as $\mu^{(\text{new})} \leftarrow \lambda\mu^{(\text{old})} + (1 - \lambda)\mu^{(\text{update})}$.

## 4  Experiments

In this section, we evaluate the performance of our proposed method as well as the state of the art for subset selection on synthetic and real sequential data. For real applications, we consider the task of summarizing instructional videos to learn the key steps of the task described in the videos. In addition to our proposed message passing (MP) algorithm, we have implemented the optimization in (12) using an ADMM framework [43], where we have relaxed the integer binary constraints to $z_{i,t} \in [0, 1]$. In practice both MP and ADMM algorithms achieve similar results. Hence, we report the performance of our method using the MP algorithm.

We compare our proposed method, Sequential Facility Location (SeqFL), with several subset selection algorithms. Since we study the performance of methods as a function of the size of the representative

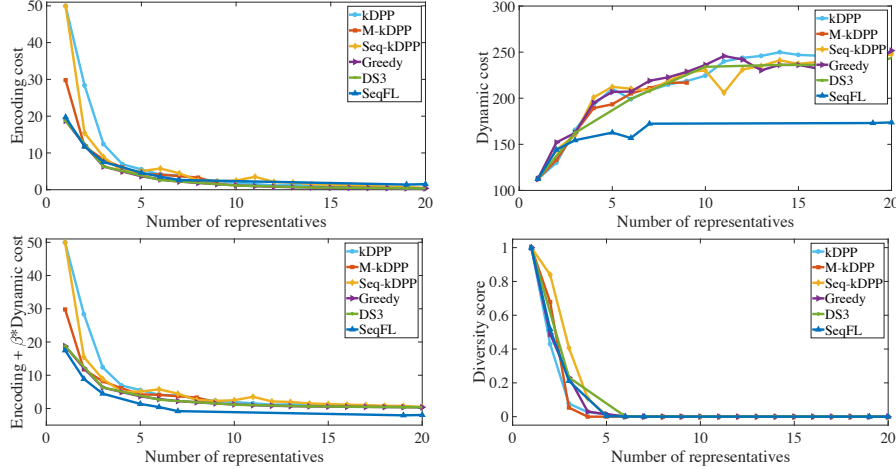

Figure 3: Encoding cost, dynamic cost, total cost and diversity score of different algorithms as a function of the number of selected representatives. The size of the source set is $M = 50$.

set, we use the fixed-size variant of DPP, called kDPP [44]. In addition to kDPP, we evaluate the performance of Markov kDPP (M-kDPP) [37], in which successive representatives are diverse among themselves and with respect to the previously selected representatives, as well as Sequential kDPP (Seq-kDPP) [3], which divides a time-series into multiple windows and successively selects diverse samples from each window conditioned on the previous window.[2] We also compare our method against DS3 [8] and the standard greedy method [28], which optimize the facility location objective function, which has no dynamic cost, via convex relaxation and greedy selection, respectively.

To compare the performance of different methods, we evaluate several costs and scores that demonstrate the effectiveness of each method in terms of encoding, diversity and dynamic compatibility of the set of selected representatives. More specifically, given dissimilarities $\{d_{i,t}\}$, the dynamic model $p_1(\cdot)$ and $p(\cdot|\cdot)$, representative set $\Lambda$, and the assignment of points to representatives $\{z_{i,t}^*\}$, we compute the encoding cost as $\sum_{t=1}^{T} \sum_{i=1}^{M} d_{i,t} z_{i,t}^*$, the dynamic cost as $-\sum_{i=1}^{M} \log p_1(\boldsymbol{x}_i) z_{i,1}^* - \sum_{t=2}^{T} \sum_{i,i'=1}^{M} \log p(\boldsymbol{x}_{i'}|\boldsymbol{x}_i) z_{i,t-1}^* z_{i',t}^*$ and the total cost as the sum of the encoding cost and the dynamic cost multiplied by $\beta$. We also compute the diversity score as $\det(\boldsymbol{K}_\Lambda)$, where $\boldsymbol{K}$ corresponds to the kernel matrix, used in DPP and its variants, and $\boldsymbol{K}_\Lambda$ denotes the submatrix of $\boldsymbol{K}$ indexed by $\Lambda$. We use Euclidean distances as dissimilarities and compute the corresponding inner-product kernel to run DPPs. Notice that the diversity score, which is the volume of the parallelotope spanned by the representatives, is what DPP methods aim to (approximately) maximize. As DPP methods only find representatives and not assignment of points, we compute $z_{i,t}^*$'s by assigning each point to the closest representative in $\Lambda$, according to the kernel.

## 4.1 Synthetic Data

To demonstrate the effectiveness of our proposed method for sequential subset selection, we generate synthetic data where for a source set $\mathbb{X}$ with $M$ items corresponding to means of $M$ Gaussians, we generate a transition probability matrix among items and an initial probability vector. We draw a sequence of length $T$ from the corresponding Markov model to form the target set $\mathbb{Y}$ and run different algorithms to generate $k$ representatives. We then compute the average encoding and transition costs as well as the diversity scores for sequences drawn from the Markov model, as a function of $k \in \{1, 2, \ldots, M\}$. In the experiments we set $M = 50, T = 100$. For a fixed $\beta$, we run SeqFL for different values of $\lambda$ to select different number of representatives.

Figure 3 illustrates the encoding, dynamic and total costs as well as the diversity scores of different methods, where for SeqFL we have set $\beta = 0.02$. Notice that our proposed method consistently obtains lower encoding, dynamic and total costs for all numbers of representatives, demonstrating its effectiveness for obtaining a sequence of informative representatives that are compatible according

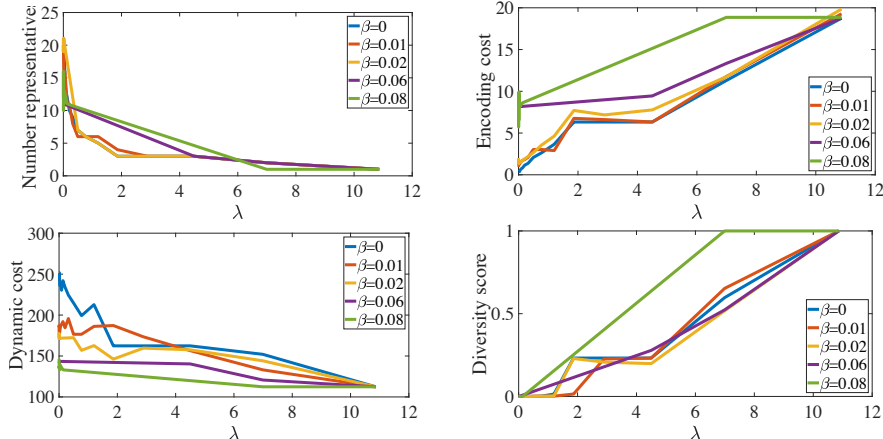

Figure 4: Number of representatives, encoding cost, dynamic cost and diversity score of our proposed method (SeqFL) as a function of the parameters $(\beta, \lambda)$.

| Task | | kDPP | M-kDPP | Seq-kDPP | DS3 | SeqFL |
|------|------|------|--------|----------|-----|-------|
| Change tire | (P, R) | (0.56, 0.50) | (0.55, 0.60) | (0.44, 0.40) | (0.56, 0.50) | (0.60, 0.60) |
| | F-score | 0.53 | 0.57 | 0.42 | 0.53 | **0.60** |
| Make coffee | (P, R) | (0.38, 0.33) | (0.50, 0.44) | (0.63, 0.56) | (0.50, 0.56) | (0.50, 0.56) |
| | F-score | 0.35 | 0.47 | **0.59** | 0.53 | 0.53 |
| CPR | (P, R) | (0.71, 0.71) | (0.71, 0.71) | (0.71, 0.71) | (0.71, 0.71) | (0.83, 0.71) |
| | F-score | 0.71 | 0.71 | 0.71 | 0.71 | **0.77** |
| Jump car | (P, R) | (0.50, 0.50) | (0.56, 0.50) | (0.56, 0.50) | (0.50, 0.50) | (0.60, 0.60) |
| | F-score | 0.50 | 0.53 | 0.53 | 0.50 | **0.60** |
| Repot plant | (P, R) | (0.57, 0.67) | (0.60, 0.50) | (0.57, 0.67) | (0.57, 0.67) | (0.80, 0.67) |
| | F-score | 0.62 | 0.55 | 0.62 | 0.62 | **0.73** |
| All tasks | (P, R) | (0.54, 0.54) | (0.58, 0.55) | (0.58, 0.57) | (0.57, 0.59) | (0.67, 0.63) |
| | F-score | 0.54 | 0.57 | 0.57 | 0.58 | **0.65** |

Table 1: Precision (P), Recall (R) and F-score for the summarization of instructional videos for five tasks.

to the underlying dynamics. It is important to notice that although our method does not maximize the diversity score, used and optimized in kDPP and its variants, it achieves slightly better diversity scores (higher is better) than kDPP and M-kDPP. Figure 4 demonstrates the effect of the parameters $(\beta, \lambda)$ on the solution of our proposed method. Notice that for a fixed $\beta$, as $\lambda$ increases, we select a smaller number of representatives, hence, the encoding cost increases. Also, for a fixed $\lambda$, as $\beta$ increases, we put more more emphasis on dynamic compatibility of representatives, hence, the dynamic cost decreases. On the other hand, the diversity score decreases for smaller $\lambda$, as we select more representatives which become more redundant. The results in Figure 4 also demonstrate the robustness of our method to the change of parameters.

## 4.2 Summarization of Instructional Videos

People learn how to perform tasks such as assembling a device or cooking a recipe, by watching instructional videos for which there often exists a large amount of videos on the internet. Summarization of instructional videos helps to learn the grammars of tasks in terms of key activities or procedures that need to be performed in order to do a certain task. On the other hand, there is a logical way in which the key actions or procedures are connected together, hence, emphasizing the importance of using the dynamic model of data when performing summarization.

We apply SeqFL to the task of summarization of intructional videos to automatically learn the sequence of key actions to perform a task. We use videos from the instructional video dataset [45], which consists of 30 instructional videos for each of five activities. The dataset also provides labels for frames which contain the main steps required to perform that task. We preprocess the videos by segmenting each video into superframes [46] and obtain features using a deep neural network that we have constructed for feature extraction for summarization tasks. We use 60% of the videos from each task as the training set to build an HMM model whose states form the source set, $\mathbb{X}$. For each of the

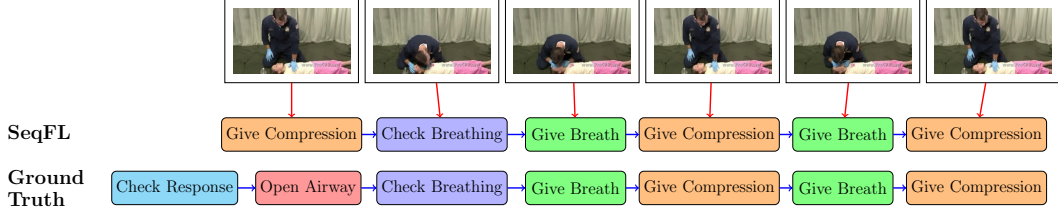

Figure 5: Ground-truth and the automatic summarization result of our method (SeqFL) for the task 'CPR'.

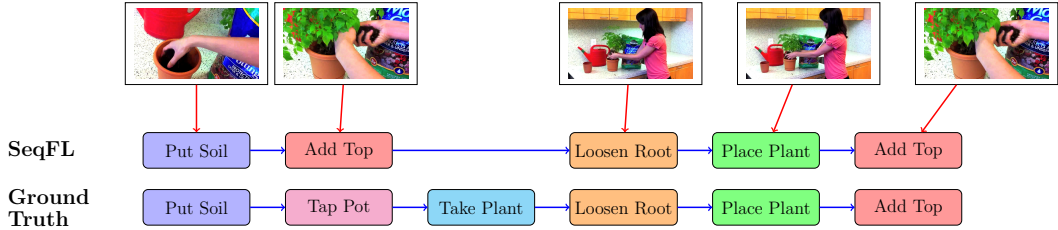

Figure 6: Ground-truth and the summarization result of our method (SeqFL) for the task 'Repot a Plant'.

40% remaining videos, we set $\mathbb{Y}$ to be the sequence of features extracted from the superframes of the video. Using the learned dynamic model, we apply our method to summarize each of these remaining videos. The summary for each video is the set of representative elements of $\mathbb{X}$, i.e., selected states from the HMM model. The assignments of representatives to superframes gives the ordering of representatives, i.e., the ordering of performing key actions.

For evaluation, we map each representative state into an action label. To do so, we use the ground-truth labels of the training videos, assigning a label to each representative state based on its five nearest neighbors in the training set. The summary for each video is an assignment of each superframe in the video to one of the representative action labels. Since each video may have shown each action performed for a different length of time, we remove consecutive repeated labels to form a list of actions performed, hence, removing the length of time each action was performed. To construct the final summary for each method for a given task, we align the lists of summary actions for all the test videos using the alignment method of [45] for several number of slots. For each method, we choose the number of HMM states and the number of slots for alignment that achieve the best performance.

Given ground-truth summaries, we compute the precision, recall and the F-score of various methods (see the supplementary materials for details). Table 1 shows the results. Notice that existing methods, which do not incorporate the dynamic of data for summarization, perform similar to each other for most tasks. In particular, the results show that the sequential diversity promoted by Seq-kDPP and M-kDPP is not sufficient for capturing the important steps of tasks. On the other hand, for most tasks and over the entire dataset, our method (SeqFL) significantly outperforms other algorithms, better producing the sequence of important steps to perform a task, thanks to the ability of our framework to incorporate the underlying dynamics of the data. Figures 5 and 6 show the ground-truth and the summaries produced by our method for the tasks 'CPR' and 'Repot a Plant', respectively. Notice that SeqFL sufficiently well captures the main steps and the sequence of steps to perform these tasks. However, for each task, SeqFL does not capture two of the ground-truth steps. We believe this can be overcome using larger datasets and more effective feature extraction methods for summarization.

## 5    Conclusions and Future Work

We developed a new framework for sequential subset selection that takes advantage of the underlying dynamic models of data, promoting to select a set of representatives that are compatible according to the dynamic models of data. By experiments on synthetic and real data, we showed the effectiveness of our method for summarization of sequential data. Our ongoing research include development of fast greedy algorithms for our sequential subset selection formulation, investigation of the theoretical guarantees of our method, as well as development of more effective summarization-based feature extraction techniques and working with larger datasets for the task of instructional data summarization.

**Acknowledgements**

This work is supported by NSF IIS-1657197 award and startup funds from the Northeastern University, College of Computer and Information Science.

## Footnotes

[1] We can also use similarities $s_{i,t}$ instead of dissimilarities, in which case we set $\phi_{\text{enc},t}(i) = \exp(s_{i,t})$.

[2]To have a fair comparison and to select a fixed number of representatives, we modify the SeqDPP method [3] and implement Seq-kDPP where $k$ representatives are chosen in each window.

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
