[Supplementary Material]

# Supplementary Materials:
# Subset Selection and Summarization
# in Sequential Data

**Ehsan Elhamifar**
Computer and Information Science College
Northeastern University
Boston, MA 02115
eelhami@ccs.neu.edu

**M. Clara De Paolis Kaluza**
Computer and Information Science College
Northeastern University
Boston, MA 02115
clara@ccs.neu.edu

## 1 Max-Sum Message Passing Algorithm

In this section, we provide the details of the message passing algorithm to optimize our proposed objective function for sequential subset selection,

$$
\begin{aligned}
\max_{\{z_{i,t}\}} \quad & \sum_{t=1}^{T}\sum_{i=1}^{M} -z_{i,t}d_{i,t} - \lambda \sum_{i=1}^{M} \| \begin{bmatrix} z_{i,1} & \cdots & z_{i,T} \end{bmatrix} \|_{\infty} + \beta(\sum_{i=1}^{M} z_{i,1} \log p_1(\boldsymbol{x}_i) \\
& + \sum_{t=2}^{T}\sum_{i,i'=1}^{M} z_{i,t-1}z_{i',t} \log p(\boldsymbol{x}_{i'}|\boldsymbol{x}_i)) \quad \text{s.t.} \quad z_{i,t} \in \{0,1\}, \quad \sum_{i=1}^{M} z_{i,t}=1, \ \forall\, i,t.
\end{aligned}
\tag{1}
$$

Since the factor graph corresponding to this objective has loops, at convergence the message passing algorithm is not guaranteed to give the maximum-a-posteriori (MAP) estimates of the binary variables $z_{i,t}$, but gives an approximation that is shown to work well in practice. Messages are pased between connected nodes and are updated iteratively until convergence. Next, we show the derivations for the message updates used in the algorithm.

For convenience of notation, we define $m_{i,i'} \triangleq \log p(\boldsymbol{x}_{i'}|\boldsymbol{x}_i)$ as the log probability of a transition from $\boldsymbol{x}_i$ at time $t$ to $\boldsymbol{x}_{i'}$ at $t+1$ and $\bar{d}_{i,t} \triangleq d_{i,t} - \log p_1(\boldsymbol{x}_i)$ if $t=1$ and $\bar{d}_{i,t} \triangleq d_{i,t}$ for all other values of $t$. This term incorporates the probability distribution of initial states into the dissimilarity term. Furthermore, we denote $\boldsymbol{z}_{i,:} \triangleq \begin{bmatrix} z_{i,1} & \cdots & z_{i,T} \end{bmatrix}^{\top}$ and $\boldsymbol{z}_{:,t} \triangleq \begin{bmatrix} z_{1,t} & \cdots & z_{M,t} \end{bmatrix}^{\top}$.

Let $\mu_{z_{i,t}\to\theta}(z_{i,t})$ denote the message that the variable $z_{i,t}$ passes to a factor node $\theta$ and $\mu_{z_{i,t}\to\theta}(z_{i,t})$ denote the message that factor node $\theta$ passes to the variable $z_{i,t}$. Let $\mathcal{N}_{z_{i,t}}$ represent the set of factor nodes with an edge connected to $z_{i,t}$ and Let $\mathcal{N}_{\theta}$ represent the set of variable nodes with an edge connected to the factor $\theta$. Then, in general, the message $\mu_{z_{i,t}\to\theta}(z_{i,t})$ is equal to the sum of all messages passed into that variable node from its neighboring factors. That is, $\mu_{z_{i,t}\to\theta}(z_{i,t}) = \sum_{\theta'\in\mathcal{N}_{z_{i,t}}\backslash\theta} \mu_{\theta'\to z_{i,t}}(z_{i,t})$. Likewise, for messages from factor nodes to variable nodes, $\mu_{\theta\to z_{i,t}}(z_{i,t}) = \max_{\sim\{z_{i,t}\}}\{\theta(z) + \sum_{z'\in\mathcal{N}_{\theta}\backslash z_{i,t}} \mu_{z'\to\theta}\}$. The notation $\sim\{z_{i,t}\}$ indicates the set of all variable nodes connected to the factor $\theta$ except $z_{i,t}$. We use these two rules to derive the message updates for each factor. Instead of keeping track of the messages evaluated at both values of the binary variables $z_{i,t}$, we keep track of the difference $\mu(1) - \mu(0)$.

Figure 1: Left: Factor graph representing the objective function in (1). Right: Messages from each factor node to a variable node for $z_{i,t}$

## Message $\sigma_{i,t}$ :

The message $\sigma_{i,t}$ sent by factor $\theta_{it}$ is associated with the encoding cost in the objective function. Since the factor $\theta_{it}$ has only one edge, the message is derived simply as follows

$$\sigma_{i,t}(z_{i,t}) = \theta_{i,t}(z_{i,t}) = -\bar{d}_{i,t} z_{i,t}$$

Since $z_{i,t}$ is a binary variable, we have $\sigma_{i,t}(z_{i,t} = 0) = 0$ and $\sigma_{i,t}(z_{i,t} = 1) = -\bar{d}_{i,t}$. Hence, defining $\sigma_{i,t} = \sigma_{i,t}(1) - \sigma_{i,t}(0)$, we have

$$\boxed{\sigma_{i,t} = -\bar{d}_{i,t}}$$

## Message $\eta_{i,t}$ :

The message $\eta_{i,t}$ sent by $\theta_t^C$ is associated with the constraint that exactly one variable $z_{i,t}$ in a column of the factor graph should be set to 1 and all the rest should be set to zero. Conceptually, this constraint enforces that each item in the set is represented by exactly one element of the source set from which representatives are chosen.

First, expand the sum of incoming messages

$$\eta_{i,t}(z_{i,t}) = \max_{k \neq i} \left\{ \theta_t^C(z_{:,t}) + \sum_{k \neq i} \mu_{z_{k,t} \to \theta_t^C}(z_{k,t}) \right\}$$

$$= \max_{k \neq i} \left\{ \theta_t^C(z_{:,t}) + \sum_{k \neq i} \left( \sigma_{k,t}(z_{k,t}) + \alpha_{k,t}(z_{k,t}) + \sum_{j=1}^{M} \gamma_{k,t;j,t+1}(z_{k,t}) + \sum_{j=1}^{M} \gamma'_{k,t;j,t-1}(z_{k,t}) \right) \right\}$$

There are two cases to consider, either $z_{i,t} = 1$ and state $i$ is chosen as a representative for $t$ or it is not chosen to represent $t$ and $z_{i,t} = 0$. In the second case, $t$ must be represented by some other state $i'$. We detail the two cases below.

$\underline{\eta_{i,t}(z_{i,t} = 1)}$: $i$ is the representative for $t$, so $z_{j,t} = 0 \ \forall k \neq i$ and $\theta_t^C = 0$

$$= 0 + \sum_{k \neq i} \left( \sigma_{k,t}(0) + \alpha_{k,t}(0) + \sum_{j=1}^{M} \gamma_{k,t;j,t+1}(0) + \sum_{j=1}^{M} \gamma'_{k,t;j,t-1}(0) \right)$$

$$= \sum_{k \neq i} \left( \alpha_{k,T}(0) + \sum_{j=1}^{M} \gamma_{k,t;j,t+1}(0) + \sum_{j=1}^{M} \gamma'_{k,t;j,t-1}(0) \right)$$

$\underline{\eta_{i,t}(z_{i,t} = 0)}$: $i$ is not representative for $t$, so its representative is some $i' \neq i$ so $z_{i',t} = 1$ and $z_{k,t} = 0 \ \forall k \neq i, i'$, also $\theta_t^C = 0$

$$
= 0 + \max_{i' \neq i} \left\{ \sum_{k \neq i', i} \left( \sigma_{k,t}(0) + \alpha_{k,t}(0) + \sum_{j=1}^{M} \gamma_{k,t;j,t+1}(0) + \sum_{j=1}^{M} \gamma'_{k,t;j,t-1}(0) \right) \right.
$$
$$
\left. + \sigma_{i',t}(1) + \alpha_{i',t}(1) + \sum_{j=1}^{M} \gamma_{i',t;j,t+1}(1) + \sum_{j=1}^{M} \gamma'_{i',t;j,t-1}(1) \right\}
$$
$$
= \max_{i' \neq i} \left\{ \sum_{k \neq i', i} \left( \alpha_{k,t}(0) + \sum_{j=1}^{M} \gamma_{k,t;j,t+1}(0) + \sum_{j=1}^{M} \gamma'_{k,t;j,t-1}(0) \right) \right.
$$
$$
\left. - \bar{d}_{i',t} + \alpha_{i',t}(1) + \sum_{j=1}^{M} \gamma_{i',t;j,t+1}(1) + \sum_{j=1}^{M} \gamma'_{i',t;j,t-1}(1) \right\}
$$

$\underline{\eta_{i,t} = \eta_{i,t}(1) - \eta_{i,t}(0)}$ :

$$
= \sum_{k \neq i} \left( \alpha_{k,t}(0) + \sum_{j=1}^{M} \gamma_{k,t;j,t+1}(0) + \sum_{j=1}^{M} \gamma'_{k,t;j,t-1}(0) \right)
$$
$$
- \max_{i' \neq i} \left\{ \sum_{k \neq i', i} \left( \alpha_{k,t}(0) + \sum_{j=1}^{M} \gamma_{k,t;j,t+1}(0) + \sum_{j=1}^{M} \gamma'_{k,t;j,t-1}(0) \right) \right.
$$
$$
\left. - \bar{d}_{i',t} + \alpha_{i',t}(1) + \sum_{j=1}^{M} \gamma_{i',t;j,t+1}(1) + \sum_{j=1}^{M} \gamma'_{i',t;j,t-1}(1) \right\}
$$
$$
= - \max_{i' \neq i} \left\{ \sum_{k \neq i', i} \left( \alpha_{k,t}(0) + \sum_{j=1}^{M} \gamma_{k,t;j,t+1}(0) + \sum_{j=1}^{M} \gamma'_{k,t;j,t-1}(0) \right) \right.
$$
$$
- \bar{d}_{i',t} + \alpha_{i',t}(1) + \sum_{j=1}^{M} \gamma_{i',t;j,t+1}(1) + \sum_{j=1}^{M} \gamma'_{i',t;j,t-1}(1)
$$
$$
\left. - \sum_{k \neq i} \left( \alpha_{k,t}(0) + \sum_{j=1}^{M} \gamma_{k,t;j,t+1}(0) + \sum_{j=1}^{M} \gamma'_{k,t;j,t-1}(0) \right) \right\}
$$

Now, split up last term into a sum over $k \neq i', i$ and the terms with $k = i'$

$$
= - \max_{i' \neq i} \left\{ \sum_{k \neq i', i} \left( \alpha_{k,t}(0) + \sum_{j=1}^{M} \gamma_{k,t;j,t+1}(0) + \sum_{j=1}^{M} \gamma'_{k,t;j,t-1}(0) \right) - \bar{d}_{i',t} + \alpha_{i',t}(1) + \sum_{j=1}^{M} \gamma_{i',t;j,t+1}(1) \right.
$$
$$
+ \sum_{j=1}^{M} \gamma'_{i',t;j,t-1}(1) - \alpha_{i',t}(0) - \sum_{j=1}^{M} \gamma_{i',t;j,t+1}(0) - \sum_{j=1}^{M} \gamma'_{i',t;j,t-1}(0)
$$
$$
\left. - \sum_{k \neq i', i} \left( \alpha_{k,t}(0) + \sum_{j=1}^{M} \gamma_{k,t;j,t+1}(0) + \sum_{j=1}^{M} \gamma'_{k,t;j,t-1}(0) \right) \right\}
$$

$$= -\max_{i' \neq i} \left\{ -\bar{d}_{i',t} + \alpha_{i',t}(1) + \sum_{j=1}^{M} \gamma_{i',t;j,t+1}(1) + \sum_{j=1}^{M} \gamma'_{i',t;j,t-1}(1) - \alpha_{i',t}(0) \right.$$

$$\left. - \sum_{j=1}^{M} \gamma_{i',t;j,t+1}(0) - \sum_{j=1}^{M} \gamma'_{i',t;j,t-1}(0) \right\}$$

$$= -\max_{i' \neq i} \left\{ -\bar{d}_{i',t} + (\alpha_{i',t}(1) - \alpha_{i',t}(0)) + \sum_{j=1}^{M} \left[ \gamma_{i',t;j,t+1}(1) - \gamma_{i',t;j,t+1}(0) \right] \right.$$

$$\left. + \sum_{j=1}^{M} \left[ \gamma'_{i',t;j,t-1}(1) - \gamma'_{i',t;j,t-1}(0) \right] \right\}$$

Since we define messages as the difference of the message evaluated at both values of the binary variables, $\alpha_{i',t} = \alpha_{i',t}(1) - \alpha_{i',t}(0)$ and likewise for the messages $\gamma_{i',t;j,t+1}$ and $\gamma'_{i',t;j,t-1}$. We substitute the differences to derive the final message update rule for message $\eta_{i,t}$.

$$\boxed{\eta_{i,t} = -\max_{k \neq i',i} \left\{ \alpha_{i',t} - \bar{d}_{i',t} + \sum_{j=1}^{M} \gamma_{i',t;j,t+1} + \sum_{j=1}^{M} \gamma'_{i',t;j,t-1} \right\}}$$

**Message $\alpha_{i,t}$ :**

The message $\alpha_{i,t}$ is sent by $\theta_i^R$ and is associated with the cardinality factor. This term controls the number of representative that are chosen using the parameter $\lambda$ since for each state that is chosen as a representative, we must "pay" $\lambda$ in the objective function. Using this idea, we can consider two cases for the state $i$ associated with $\theta_i^R$: either it is chosen as a representative and the factor evaluates to $-\lambda$, or it is not and $\theta_i^R = 0$. First, we expand the sum of incoming messages.

$$\alpha_{i,t}(z_{i,t}) = \max_{k \neq t} \left\{ \theta_i^R(z_{i,:}) + \sum_{k \neq T} \mu_{z_{i,k} \to \theta_i^R}(z_{i,k}) \right\}$$

$$= \max_{k \neq t} \left\{ \theta_i^R(z_{i,:}) + \sum_{k \neq t} \left( \sigma_{ik}(z_{i,k}) + \eta_{i,k}(z_{i,k}) + \sum_{j=1}^{M} \left[ \gamma_{i,k;j,k+1}(z_{i,k}) + \gamma'_{i,k;j,k-1}(z_{i,k}) \right] \right) \right\}$$

$$= \theta_i^R(z_{i,:}) + \sum_{k \neq t} \max_{k \neq t} \left\{ \sigma_{i,k}(z_{i,k}) + \eta_{i,k}(z_{i,k}) + \sum_{j=1}^{M} \left[ \gamma_{i,k;j,k+1}(z_{i,k}) + \gamma'_{i,k;j,k-1}(z_{i,k}) \right] \right\}$$

$\underline{\alpha_{i,t}(z_{i,t} = 1)}$: $i$ is chosen as a representative for at least one point ($t$), so $\theta_i^R = -\lambda$

$$= -\lambda + \sum_{k \neq,} \max \left\{ \sigma_{i,k}(0) + \eta_{i,k}(0) + \sum_{j=1}^{M} \left[ \gamma_{,ik;j,k+1}(0) + \gamma'_{i,k;j,k-1}(0) \right] ; \right.$$

$$\left. \sigma_{i,k}(1) + \eta_{i,k}(1) + \sum_{j=1}^{M} \left[ \gamma_{i,k;j,k+1}(1) + \gamma'_{i,k;j,k-1}(1) \right] \right\}$$

$$= -\lambda + \sum_{k \neq l} \max \left\{ 0; -\bar{d}_{i,k} + \eta_{i,k} + \sum_{j=1}^{M} \left[ \gamma_{i,k;j,k+1} + \gamma'_{i,k;j,k-1} \right] \right\}$$

$$+ \sum_{k \neq t} \left( \eta_{i,k}(0) + \sum_{j=1}^{M} \left[ \gamma_{i,k;j,k+1}(0) + \gamma'_{i,k;j,k-1}(0) \right] \right)$$

$\alpha_{i,t}(z_{i,t}=0)$: $i$ is not representative for $t$, but it may or may not be selected as a representative for some other point(s)

$$= \max \left\{ 0 + \sum_{k \neq t} \sigma_{i,k}(0) + \eta_{i,k}(0) + \sum_{j=1}^{M} \left[ \gamma_{i,k;j,k+1}(0) + \gamma'_{i,k;j,k-1}(0) \right] ; \right.$$

$$- \lambda + \sum_{k \neq t} \max \left\{ \sigma_{i,k}(0) + \eta_{i,k}(0) + \sum_{j=1}^{M} \left[ \gamma_{i,k;j,k+1}(0) + \gamma'_{i,k;j,k-1}(0) \right] ; \right.$$

$$\left. \left. \sigma_{i,k}(1) + \eta_{i,k}(1) + \sum_{j=1}^{M} \left[ \gamma_{i,k;j,k+1}(1) + \gamma'_{i,k;j,k-1}(1) \right] \right\} \max \right\}$$

$$= \left\{ 0 ; -\lambda + \sum_{k \neq t} \max \left\{ 0 ; -\bar{d}_{i,k} + \eta_{i,k} + \sum_{j=1}^{M} \left[ \gamma_{i,k;j,k+1} + \gamma'_{i,k;j,k-1} \right] \right\} \right\}$$

$$+ \sum_{k \neq t} \left( \eta_{i,k}(0) + \sum_{j=1}^{M} \left[ \gamma_{i,k;j,k+1}(0) + \gamma'_{i,k;j,k-1}(0) \right] \right)$$

For convenience of notation, define the following quantities:

$$\psi \triangleq \sum_{k \neq t} \left( \eta_{i,k}(0) + \sum_{j=1}^{M} \left[ \gamma_{i,k;j,k+1}(0) + \gamma'_{i,k;j,k-1}(0) \right] \right)$$

$$\phi \triangleq -\lambda + \sum_{k \neq t} \max \left\{ 0 ; -\bar{d}_{i,k} + \eta_{i,k} + \sum_{j=1}^{M} \left[ \gamma_{i,k;j,k+1} + \gamma'_{i,k;j,k-1} \right] \right\}$$

so that $\alpha_{i,t}(1) = \phi + \psi$ and $\alpha_{i,t}(0) = \max\{0, \phi\} + \psi$.

$\alpha_{i,t} = \alpha_{i,t}(1) - \alpha_{i,t}(0)$ :

$$= \phi + \psi - \max\{0, \phi\} - \psi$$
$$= \phi - \max\{0, \phi\} = \phi + \min\{0, -\phi\} = \min\{0, \phi\}$$

$$\boxed{\alpha_{i,t} = \min \left\{ 0 ; -\lambda + \sum_{k \neq t} \max \left\{ 0 ; -\bar{d}_{i,k} + \eta_{i,k} + \sum_{j=1}^{M} \left[ \gamma_{i,k;j,k+1} + \gamma'_{i,k;j,k-1} \right] \right\} \right\}}$$

**Messages** $\gamma_{i,t;jl+1}$ :

Messages $\gamma_{i,t;j,t+1}$ are associated with forward transitions between the representative of $t$ and of $t+1$. Note that for each $(i,t)$, there are $M$ messages $\gamma_{i,t;j,t+1}$, one for each $j = 1, \ldots, M$. That is, there are $M$ possible representatives at $t+1$ to transition to from the representative at $t$.

$$\gamma_{i,t;j,t+1} = \theta_{i,t;j,t+1}^{M}(z_{i,t}, z_{j,t+1}) + \mu_{z_{j,t+1} \to \theta_{i,t;j,t+1}^{M}}(z_{j,t+1})$$

$$= m_{ij} z_{i,t} z_{j,t+1} + \sigma_{j,t+1}(z_{j,t+1}) + \alpha_{j,t+1}(z_{j,t+1}) + \eta_{j,t+1}(z_{j,t+1})$$

$$+ \sum_{m=1}^{M} \gamma_{j,t+1;m,t+1}(z_{j,t+1}) + \sum_{m \neq i} \gamma'_{j,t+1;m,t}(z_{j,t+1})$$

$\gamma_{i,t;j,t+1}(z_{i,t}=1)$: $i$ is the representative for $t$. For the representative $j$ considered by this message, $z_{j,t+1}$ may equal 0 or 1. If $z_{j,t+1} = 1$, $j$ is the representative for $t+1$ and there is a transition from $i$

to $j$ and $\theta^M_{i,t;j,t+1}(z_{i,t}, z_{j,t+1}) = m_{ij}$, otherwise $z_{j,t+1} = 0$, $j$ is not the representative for $t+1$ and $\theta^M_{i,t;j,t+1}(z_{i,t}, z_{j,t+1}) = 0$.

$$= \max \left\{ m_{ij} + \sigma_{j,t+1}(1) + \alpha_{j,t+1}(1) + \eta_{j,t+1}(1) + \sum_{k=1}^{M} \gamma_{j,t+1;k,t+2}(1) + \sum_{k \neq i} \gamma'_{j,t+1;k,t}(1); \right.$$

$$\left. 0 + \sigma_{j,t+1}(0) + \alpha_{j,t+1}(0) + \eta_{j,t+1}(0) + \sum_{k=1}^{M} \gamma_{j,t+1;k,t+2}(0) + \sum_{k \neq i} \gamma'_{j,t+1;k,t}(0) \right\}$$

$$= \max \left\{ m_{ij} - \bar{d}_{j,t+1} + \alpha_{j,t+1}(1) + \eta_{j,t+1}(1) + \sum_{k=1}^{M} \gamma_{j,t+1;k,t+2}(1) + \sum_{k \neq i} \gamma'_{j,t+1;k,t}(1); \right.$$

$$\left. + \alpha_{j,t+1}(0) + \eta_{j,t+1}(0) + \sum_{k=1}^{M} \gamma_{j,t+1;k,t+2}(0) + \sum_{k \neq i} \gamma'_{j,t+1;k,t}(0) \right\}$$

$$= \max \left\{ m_{ij} - \bar{d}_{j,t+1} + \alpha_{j,t+1} + \eta_{j,t+1} + \sum_{k=1}^{M} \gamma_{j,t+1;k,t+2} + \sum_{k \neq i} \gamma'_{j,t+1;k,t}; 0 \right\}$$

$$+ \alpha_{j,t+1}(0) + \eta_{j,t+1}(0) + \sum_{k=1}^{M} \gamma_{j,t+1;k,t+2}(0) + \sum_{k \neq i} \gamma'_{j,t+1;k,t}(0)$$

$\underline{\gamma_{i,t;j,t+1}(z_{i,t} = 0)}$: $i$ is not representative for $t$. For the representative $j$ considered by this message, $z_{j,t+1}$ may equal 0 or 1. Regardless, since $i$ is not representative for $t$, $z_{i,t} = 0$ and $\theta^M_{i,t;j,t+1}(z_{i,t}, z_{j,t+1}) = 0$

$$= \max \left\{ 0 + \sigma_{j,t+1}(1) + \alpha_{j,t+1}(1) + \eta_{j,t+1}(1) + \sum_{k=1}^{M} \gamma_{j,t+1;k,t+2}(1) + \sum_{k \neq i} \gamma'_{j,t+1;k,t}(1); \right.$$

$$\left. 0 + \sigma_{j,t+1}(0) + \alpha_{j,t+1}(0) + \eta_{j,t+1}(0) + \sum_{k=1}^{M} \gamma_{j,t+1;k,t+2}(0) + \sum_{k \neq i} \gamma'_{j,t+1;k,t}(0) \right\}$$

$$= \max \left\{ -\bar{d}_{j,t+1} + \alpha_{j,t+1}(1) + \eta_{j,t+1}(1) + \sum_{k=1}^{M} \gamma_{j,t+1;k,t+2}(1) + \sum_{k \neq i} \gamma'_{j,t+1;k,t}(1); \right.$$

$$\left. + \alpha_{j,t+1}(0) + \eta_{j,t+1}(0) + \sum_{k=1}^{M} \gamma_{j,t+1;k,t+2}(0) + \sum_{k \neq i} \gamma'_{j,t+1;k,t}(0) \right\}$$

$$= \max \left\{ -\bar{d}_{j,t+1} + \alpha_{j,t+1} + \eta_{j,t+1} + \sum_{k=1}^{M} \gamma_{j,t+1;k,t+2} + \sum_{k \neq i} \gamma'_{j,t+1;k,t}; 0 \right\}$$

$$+ \alpha_{j,t+1}(0) + \eta_{j,t+1}(0) + \sum_{k=1}^{M} \gamma_{j,t+1;k,t+2}(0) + \sum_{k \neq i} \gamma'_{j,t+1;k,t}(0)$$

$\underline{\gamma_{i,t;j,t+1} = \gamma_{i,t;j,t+1}(1) - \gamma_{i,t;j,t+1}(0)}$ :

Define

$$\boxed{\rho \overset{\triangle}{=} -\bar{d}_{j,t+1} + \alpha_{j,t+1} + \eta_{j,t+1} + \sum_{k=1}^{M} \gamma_{j,t+1;k,t+2} + \sum_{k \neq i} \gamma'_{j,t+1;k,t}}$$

$$\gamma_{i,t;j,t+1}(1) - \gamma_{i,t;j,t+1}(0)$$

$$= \max\{m_{ij} + \rho; 0\} + \alpha_{j,t+1}(0) + \eta_{j,t+1}(0) + \sum_{k=1}^{M} \gamma_{j,t+1;k,t+2}(0) + \sum_{k\neq i} \gamma'_{j,t+1;k,t}(0)$$

$$- \max\{\rho; 0\} - \alpha_{jl+1}(0) - \eta_{j,t+1}(0) - \sum_{k=1}^{M} \gamma_{j,t+1;k,t+2}(0) - \sum_{k\neq i} \gamma'_{j,t+1;k,t}(0)$$

$$= \max\{m_{ij} + \rho; 0\} - \max\{\rho; 0\}$$

$$\boxed{\gamma_{i,t;jl+1} = \max\{m_{ij} + \rho; 0\} - \max\{\rho; 0\}}$$

**Messages $\gamma'_{i,t;j,t-1}$ :**

Messages $\gamma'_{i,t;j,t-1}$ are associated with transitions between the representative at $t-1$ and $t$. Note that for each $(i,t)$, there are $M$ messages $\gamma'_{i,t;j,t-1}$, one for each $j = 1, \ldots, M$. That is, there are $M$ possible representatives at $t-1$ from which to transition at $t$.

$$\gamma'_{i,t;j,t-1}(z_{i,t}) = \theta^M_{j,t-1;i,t}(z_{j,t-1}, z_{i,t}) + \mu_{z_{j,t-1} \to \theta^M_{j,t-1;i,t}}(z_{j,t-1})$$

$$= m_{ji}z_{j,t-1}z_{i,t} + \sigma_{j,t-1}(z_{j,t-1}) + \alpha_{j,t-1}(z_{j,t-1}) + \eta_{j,t-1}(z_{j,t-1})$$

$$+ \sum_{k\neq j} \gamma_{j,t-1;k,t}(z_{j,t-1}) + \sum_{k=1}^{M} \gamma'_{j,t-1;k,t-2}(z_{j,t-1})$$

$\gamma'_{i,t;j,t-1}(z_{i,t} = 1)$: $i$ is the representative for $t$. For the representative $j$ considered by this message, $z_{j,t-1}$ may equal 0 or 1. If $z_{j,t-1} = 1$, there is a transition from representative $j$ to representative $i$ and $\theta^M_{j,t-1;i,t}(z_{j,t-1}, z_{i,t}) = m_{ji}$, otherwise $z_{j,t-1} = 0$ and $\theta^M_{j,t-1;i,t}(z_{j,t-1}, z_{i,t}) = 0$.

$$= \max \left\{ m_{ji} + \sigma_{j,t-1}(1) + \alpha_{j,t-1}(1) + \eta_{j,t-1}(1) + \sum_{k\neq j} \gamma_{j,t-1;k,t}(1) + \sum_{k=1}^{M} \gamma'_{j,t-1;k,t-2}(1); \right.$$

$$\left. 0 + \sigma_{j,t-1}(0) + \alpha_{j,t-1}(0) + \eta_{j,t-1}(0) + \sum_{k\neq j} \gamma_{j,t-1;k,t}(0) + \sum_{k=1}^{M} \gamma'_{j,t-1;k,t-2}(0) \right\}$$

$$= \max \left\{ m_{ji} - \bar{d}_{j,t-1} + \alpha_{j,t-1}(1) + \eta_{j,t-1}(1) + \sum_{k\neq j} \gamma_{j,t-1;k,t}(1) + \sum_{k=1}^{M} \gamma'_{j,t-1;k,t-2}(1); \right.$$

$$\left. + \alpha_{j,t-1}(0) + \eta_{j,t-1}(0) + \sum_{k\neq j} \gamma_{j,t-1;k,t}(0) + \sum_{k=1}^{M} \gamma'_{j,t-1;k,t-2}(0) \right\}$$

$$= \max \left\{ m_{ji} - \bar{d}_{j,t-1} + \alpha_{j,t-1} + \eta_{j,t-1} + \sum_{k\neq j} \gamma_{j,t-1;k,t} + \sum_{k=1}^{M} \gamma'_{j,t-1;k,t-2}; 0 \right\}$$

$$+ \alpha_{j,t-1}(0) + \eta_{j,t-1}(0) + \sum_{k\neq j} \gamma_{j,t-1;k,t}(0) + \sum_{k=1}^{M} \gamma'_{j,t-1;k,t-2}(0)$$

$\overline{\gamma'_{i,t;j,t-1}(z_{i,t}=0)}$: $i$ is not representative for $t$. For this fixed $j$, $z_{j,t-1}$ may equal 0 or 1. Regardless, since $z_{i,t}=0$), $\theta^M_{j,t-1;i,t}(z_{j,t-1}, z_{i,t}) = 0$

$$
= \max \left\{ 0 + \sigma_{j,t-1}(1) + \alpha_{j,t-1}(1) + \eta_{j,t-1}(1) + \sum_{k \neq j} \gamma_{j,t-1;k,t}(1) + \sum_{k=1}^{M} \gamma'_{j,t-1;k,t-2}(1); \right.
$$
$$
\left. 0 + \sigma_{j,t-1}(0) + \alpha_{j,t-1}(0) + \eta_{j,t-1}(0) + \sum_{k \neq j} \gamma_{j,t-1;k,t}(0) + \sum_{k=1}^{M} \gamma'_{j,t-1;k,t-2}(0) \right\}
$$

$$
= \max \left\{ -\bar{d}_{j,t-1} + \alpha_{j,t-1}(1) + \eta_{j,t-1}(1) + \sum_{k \neq j} \gamma_{j,t-1;k,t}(1) + \sum_{k=1}^{M} \gamma'_{j,t-1;k,t-2}(1); \right.
$$
$$
\left. + \alpha_{j,t-1}(0) + \eta_{j,t-1}(0) + \sum_{k \neq j} \gamma_{j,t-1;k,t}(0) + \sum_{k=1}^{M} \gamma'_{j,t-1;k,t-2}(0) \right\}
$$

$$
= \max \left\{ -\bar{d}_{j,t-1} + \alpha_{j,t-1} + \eta_{j,t-1} + \sum_{k \neq j} \gamma_{j,t-1;k,t} + \sum_{k=1}^{M} \gamma'_{j,t-1;k,t-2}; 0 \right\}
$$
$$
+ \alpha_{j,t-1}(0) + \eta_{j,t-1}(0) + \sum_{k \neq j} \gamma_{j,t-1;k,t}(0) + \sum_{k=1}^{M} \gamma'_{j,t-1;k,t-2}(0)
$$

$\overline{\gamma'_{i,t;j,t-1} = \gamma'_{i,t;j,t-1}(1) - \gamma'_{i,t;j,t-1}(0)}$ :

Define

$$
\boxed{\rho' \triangleq -\bar{d}_{j,t-1} + \alpha_{j,t-1} + \eta_{j,t-1} + \sum_{k \neq j} \gamma_{j,t-1;k,t} + \sum_{k=1}^{M} \gamma'_{j,t-1;k,t-2}}
$$

$$
\gamma'_{i,t;j,t-1}(1) - \gamma'_{i,t;j,t-1}(0)
$$
$$
= \max\{m_{ji} + \rho'; 0\} + \alpha_{j,t-1}(0) + \eta_{j,t-1}(0) + \sum_{k \neq j} \gamma_{j,t-1;k,t}(0) + \sum_{k=1}^{M} \gamma'_{j,t-1;k,t-2}(0)
$$
$$
- \max\{\rho'; 0\} - \alpha_{j,t-1}(0) - \eta_{j,t'1}(0) - \sum_{k \neq j} \gamma_{j,t-1;k,t}(0) - \sum_{k=1}^{M} \gamma'_{j,t-1;k,t-2}(0)
$$
$$
= \max\{m_{ji} + \rho'; 0\} - \max\{\rho'; 0\}
$$

$$
\boxed{\gamma'_{i,t;j,t-1} = \max\{m_{ji} + \rho'; 0\} - \max\{\rho'; 0\}}
$$

## 2 Instructional Video Summarization Experiments

### 2.1 Generating Ground-Truth Summaries

The dataset in [1] provides a ground truth constructed by an agreement among the authors and do not contain repetition of steps, even if a task requires repeating an action to complete the task. Thus, we generate new ground truths from the labels provided for the videos in the dataset. To do so, we extract all labels and form a sequence of labels so that no subsequent labels are the same. By doing so we remove the notion of the length of time an action should take, but preserve the order and any repetition of steps required to complete a task. Once we have these sequences of labels for each video corresponding to a task, we use the alignment scheme proposed by [1] to form a single ground truth for the task.

### 2.2 Generating Automatic Summaries

For each method, we generate a summary for each individual test video. We then join the individual summaries to form the final, single summary for each task. To do so, we map the representative states selected by each method into actions in the ground truth by using the labels of the five nearest neighbors in the training set to each state. This provides individual video summaries in the form of a list of actions found in the ground truth. We then align these sequences of actions using the alignment method proposed in [1]. This method aligns multiple sequences into $L$ slots, placing each action in a sequence into a slot. If two sequences are exactly the same, they can be aligned perfectly into $L$ slots, where $L$ is the length of the sequences and each slot contains exactly the actions in the two sequences. However, if the sequences differ, more slots are required so that actions placed into the same slots are equal. After alignment, we generate the final summary by majority vote. For each slot, each sequence (video) will contribute an action from its summary or will be blank for that video. Thus, the slots with the most actions aligned into them are those that have the most agreement across videos. We take the top $k$ slots as the final summary and consider $k$ as close to the size of the ground truth as possible. The number of slots required for the alignment is parameter of the optimization.

### 2.3 Evaluation

For our experiments, we evaluate the alignment over a range of number of slots and for each method report the alignment that gives best results in terms of F-score. To evaluate summaries, we align each summary obtained by each method with the ground truth, again following the alignment procedure of [1]. We grow the number of alignment slots until there are no empty slots. That is, each slot in the alignment has an action either from the proposed summary or the ground truth, or both if they are matched. We calculate precision as the number of matched actions in the final alignment divided by the length of the proposed summary and calculate the recall as the number of matched actions divided by the number of the ground truth actions. Finally, the F score is the harmonic mean of the precision and recall measures, i.e.,

$$F = 2 \cdot \frac{\text{precision} \cdot \text{recall}}{\text{precision} + \text{recall}}.$$

## References

[1] J.-B. Alayrac, P. Bojanowski, N. Agrawal, I. Laptev, J. Sivic, and S. Lacoste-Julien, "Unsupervised learning from narrated instruction videos," in *Computer Vision and Pattern Recognition (CVPR)*, 2016.