[Reviews · NeurIPS 2017]

Reviewer 1



This paper presents a framework for sequential data summarization. The solution is based on the maximization of a potential function with three components: the first one enforces the selection of the best set, the second one promotes the sparsity of the selected set, and the third one enforces the selected set to obey a first-order Markovian model. The effect of the last two components are controlled by parameters. The resulting optimization problem is not convex and the authors propose a message passing algorithm that seems to work well in practice using a memory trick for slowing down the convergence. The proposed method is an alternative to the sequential extensions of the Determinantal Point Process (DPP) and compares favorably with it. Quality The technical formulation of the method seems sound and correct. The method is compared favorably using synthetic and real data with sequential and non-sequential summarization methods. Clarity The paper is well written and easy to understand. More details about the values of the parameters in the real experiments and an explanation about the low value of the parameter that enforces the Markovian model would be welcome. Originality As noted previously, the method is an original alternative to the sequential extensions of DPP. Significance The method represents a nice tool for the summarization of video and behavioral data.

Reviewer 2



The paper consider a new problem of subset selection from set X that represents objects from sequence Y with additional constraint that the selected subset should comply with underlaying dynamic model of set X. The problem seems interesting to me and worth investigation. The novel problem boils down to maximization problem (11). The authors suggest a new message-passing scheme for the problem. The problem can be seen as a multilabel energy minimization problem (T nodes, M labels, chain structure with additional global potential -log(\Phi_{card}(x_{r_1}, \ldots, x_{r_T}))) written in overcomplete representation form [23]. The problem is well known in energy minimization community as energy minimization with label costs [2]. It is shown that global potential -log(\Phi_{card}(x_{r_1}, \ldots, x_{r_T})) can be represented as a set of pairwise submodular potentials with auxiliary variables [2, 3]. Therefore, the problem (11) can be seen as multilabel energy minimization with pairwise potentials only. There are tons of methods including message passing based to solve the problem approximately (TRW-S, LBP, alpha-beta swap, etc). While they may work worse than the proposed message passing, the thoughtful comparison should be performed. I believe that the paper in its current form cannot be accepted. Although the considered problem is interesting, the related work section should include [1,2,3]. Also, different baselines for experimental evaluation should be used (other optimization schemes for (11)), as it is clear from the experiments that DPP based methods do not suit for the problem. I think that the improved version may have a great impact. typos: - (8) \beta was missed - L229 more more -> more - L236 jumping, jumping -> jumping [1] Wainwright, M.J., Jordan, M.I.: Graphical models, exponential families, and variational inference. Foundations and Trends in Machine Learning, 2008 [2] Delong, Andrew, et al. "Fast approximate energy minimization with label costs." International journal of computer vision 96.1 (2012): 1-27. [3] M. A. Fischler and R. C. Bolles. Random sample consensus: a paradigm for model fitting with applications to image analysis and automated cartography. Communications of the ACM, 24(6):381–395, 1981.

Reviewer 3



This paper proposes a subset selection model for sequential data. Compared to prior work, this model takes into account the underlying dynamics in the sequential data by incorporating a dynamic term in the modelling of the potential function. Maximising the dynamic potential term pushes the selected sequence of representatives to follow (match) the dynamics in the target set. The paper is well motivated while the idea is straightforward. The experimental results shows the significant improvement over the baseline models, however, the datasets used are not hard enough to be convincing. Pros: 1. This paper proposes a feasible way to model the dynamics in the sequential data, which hasn't been pay much attention to in prior work. 2. Although the optimization of the model is non-convex, the paper provides a message-passing based algorithm to tackle it. Cons: 1. The dynamic potential only considers the first-order Markov Model, which limits the modelling power of dynamics. 2. Since the three potential terms are relatively independent to each other, hence the performance is sensitive to the tuning of the hyper-parameters (\Beta and \gamma). 3. The experiments show the significantly better performance over the other baseline models, nevertheless, more quantitative experiments on harder dataset are expected.